# Assessing the Reliability of the Sexual Violence Questionnaire in Sport among Spanish-Speaking Athletes

**DOI:** 10.3390/ijerph21091214

**Published:** 2024-09-16

**Authors:** Andrea Sáenz-Olmedo, Aitor Iturricastillo, Jon Brain, Luis Maria Zulaika, Oidui Usabiaga

**Affiliations:** 1Society, Sport and Physical Activity (GIKAFIT) Research Group, Department of Physical Education and Sports, Faculty of Education and Sport, University of the Basque Country (UPV/EHU), 01007 Vitoria-Gasteiz, Spain; andrea.saenzo@ehu.eus (A.S.-O.); oidui.usabiaga@ehu.eus (O.U.); 2AKTIBOki: Research Group in Physical Activity, Physical Exercise and Sport, Department of Physical Education and Sports, Faculty of Education and Sport, University of the Basque Country (UPV/EHU), 01007 Vitoria-Gasteiz, Spain; 3Physical Activity, Exercise, and Health Group, Bioaraba Health Research Institute, 01007 Vitoria-Gasteiz, Spain; 4Safeguarding Sport and Society, Centre of Expertise Care and Well-Being, Thomas More University of Applied Sciences, 2440 Antwerp, Belgium; jon.brain@thomasmore.be; 5Department of Physical and Sports Education, Faculty of Education and Sport, University of the Basque Country (UPV/EHU), 01007 Vitoria, Spain; luismzulaika@ehu.eus

**Keywords:** sexual violence, experiences, perceptions, coach–athlete relationship, sport

## Abstract

The prevalence of sexual harassment and abuse in school sport, specifically by coaches against their athletes, remains a concerning and pervasive issue. In an attempt to better understand and prevent specific coach-behaviours associated with such sexual misconduct, researchers have developed the Sexual Violence Questionnaire in Sport. While the reliability of this measurement tool has been tested in Anglo-Saxon cultural contexts, it is not known whether the questionnaire is applicable to other cultural contexts. This study aimed to analyse the internal consistency and reliability of the questionnaire on sexual harassment in sport, originally designed and developed in English. A sample of 146 (52 female, 94 male) undergraduate students from a university in the Basque Country participated in this cross-sectional study. The questionnaire was administered twice over a two-week period to assess test–retest reliability. The internal consistency of the Sexual Violence Questionnaire in Sport was high, with Cronbach’s alpha values of 0.891 for perceptions and 0.813 for experiences across all participants. Gender-specific analysis showed similar reliability, with females having slightly lower alpha values for perceptions. Although significant differences were observed between the test and the retest on eight perception items and one experience item, Cohen’s kappa analysis indicated agreement on all items; however, some of them were low (e.g., 0.13). In conclusion, the study highlights the questionnaire’s overall reliability and suggests its effectiveness as a tool for measuring sexual violence in sport within the Spanish context. Nonetheless, the findings of this study underscore the need for further research to enhance the instrument’s stability and to better understand gender differences in perceptions and experiences of sexual violence in sport contexts.

## 1. Introduction

The prevalence of sexual harassment and abuse in sport has been extensively documented and reported in academic research [1]. Young athletes, in particular, are at significant risk of being victims of sexual violence at this age; unacceptable behaviour is often accepted as part of sport culture [2]. In a study involving 10,000 athletes from six European countries, it was found that 35% and 20% of athletes had experienced non-contact sexual violence and contact violence, respectively, before the age of 18 [3]. The identification of sexual violence behaviours is hindered by a number of factors, including the normalisation of physical contact, the intensity of the relationships between coaching staff and athletes, and the diversity of leadership styles [4]. It is also important to note that sexual violence in sport can manifest in various forms, including overt sexual behaviour, such as rape, and less obvious forms, such as non-contact verbal sexual harassment [3]. Understanding the relationship between sport participation and sexual violence behaviours could be crucial for prevention, as recent studies have identified risk and protective factors associated with sexual violence in sport [5,6]. 

Interpersonal sexual violence in sport can be perpetrated by other athletes or by adults in positions of power in the sport context. Coaches are among those who are in authoritative roles and are perceived as dominant figures by young athletes. Due to their complex and complicated relationships with their athletes, coaches can indeed be common perpetrators [1,7]. Coaches are in a position of trust and power over young athletes [2,8] and play a vital role in the lives of young athletes. Indeed, they can also provide young athletes with supportive relationships based on trust and closeness. However, close relationships and the creation of strong emotional bonds can be a trigger for sexual violence [9,10]. One aspect of risk in the coach–athlete relationship is the potential for role ambiguity, where appropriate and inappropriate behaviours are not clearly defined due to the time spent by both parties in and out of the sporting environment [11,12]. Furthermore, a coach’s role frequently entails physical contact in ways that can be instructional (e.g., teaching a start from a swimming pool) or non-educational (e.g., sharing space during a training camp or championship) [4,13,14]. The literature additionally identifies other characteristics of the coach–athlete relationship that can significantly increase the likelihood of sexual violence between a coach and athlete, including athlete dependency, age differences, and the drive for success [15,16,17]. 

Due to the challenges and the grey areas in determining what constitutes sexual harassment and abuse in sport, it is suggested that it is essential to determine what appropriate and inappropriate coaching behaviours should look like or take place within the coach–athlete relationship [15]. One of the first retrospective questionnaires on sexual violence in sport, called the Sexual Violence Questionnaire in Sport, was designed and developed to identify different types of coach behaviours likely to lead to sexual violence towards athletes [16]. Specifically, the questionnaire was first used to measure student athletes’ perceptions and experiences of sexual harassment in sport. Although subsequent questionnaires on sexual violence in sport have been developed, the inclusion of a section specifically addressing perceptions of behaviours and the coach–athlete relationship enables us to explore the nuanced areas of violence in sports. Although other questionnaires on sexual violence in sport have subsequently been developed [17], the inclusion of a section specifically on perceptions of behaviour and the coach–athlete relationship allows us to explore nuanced areas of violence in sports. Consequently, understanding these different perceptions of sexual harassment and abuse is crucial for identifying grey areas where the athletes’ perspectives are of the utmost importance [18]. 

The cultural factor also plays a vital role in how certain athletes perceive and process sexual violence [19]. For example, existent studies have confirmed these cultural variations, concluding that Israeli students considered more behaviours as inappropriate compared to American students and that Indian athletes classified more actions as sexual violence compared to European and North American students. In contrast, the perceptions of Danish athletes were similar to those of North American students [20]. These cultural differences in perceptions and experiences demonstrate the need for research in different countries and cultural contexts, particularly in Spain. This allows safeguarding policies that align with societal norms to be adopted, thereby promoting a safe sporting environment [21].

The suitability of the questionnaire regarding sexual violence [16] was also considered after small modifications were introduced and it was translated into Catalan and Spanish [22]. This was carried out after the content was analysed in a previous pilot study with participants sharing the same characteristics. However, due to the cultural differences highlighted in other research and the fact that the questionnaire was initially developed in English and subsequently adapted for this cultural context, along with the absence of a reliability analysis for the Spanish version, this study sought to evaluate the internal consistency and reliability of the sexual harassment questionnaire in sport. This questionnaire was initially designed and developed in English [16] and later adapted to Spanish/Catalan [22]. Considering that the majority of instruments have been developed and validated in English-speaking contexts [17], by studying their reliability in a Spanish-speaking population, we ensure that the questionnaire is consistent for this specific group, avoid misinterpretations that could compromise the quality of the data, develop effective prevention policies adapted to the needs and realities of Spanish-speaking athletes, and allow for comparative studies between different countries, which is fundamental to understanding the similarities and differences of sexual violence in sport worldwide [23].

## 2. Materials and Methods

### 2.1. Participants

Out of 400 students, a total of 146 students from the Bachelor of Physical Activity and Sport Sciences (CAFYD) at the University of the Basque Country (UPV/EHU) participated in this study [female: *n* = 52 (35.6%); 20.3 ± 1.9 years; male: *n* = 94 (64.4%); 21.2 ± 2.3 years]. A convenience sample was employed, with inclusion contingent upon students being enrolled in the relevant degree programme and university. This sampling approach was driven by the imperative to raise awareness of sexual harassment and abuse in sport among these student athletes, who are and will continue to be responsible for the programmes, clubs, and sports federations of the Autonomous Community.

### 2.2. Procedure

Before administering the questionnaire, all participants were informed of the study’s objectives, research procedure, and possible risks. Their participation was voluntary, and they signed an informed consent form. The study was approved by the Ethics Committee for Human Research of the University of the Basque Country (M10/2022/110).

Regarding the translation of the questionnaire, a cultural adaptation process was conducted to ensure that the items were comprehensible and relevant in the context of Spanish-speaking participants. After the initial translation into Spanish and Basque, a pilot study was conducted with a group similar to the final sample. This pilot, which included students from the same academic program who shared similar demographic and cultural characteristics, was instrumental in identifying terms and phrases that could cause ambiguity or misinterpretation. For example, it was observed that certain terms in Basque, such as ‘pinches’ or ‘diminutives’, were not clear to all participants. These terms were reviewed and reformulated to improve comprehension, ensuring that the questionnaire was clear and accessible to all participants, thereby enhancing the validity of the data collected.

In order to analyse the reliability of the Sexual Violence Questionnaire in Sport, a cross-sectional study design was carried out. The questionnaires were administered in person during the month of April 2022. Responses were collected 2 weeks apart between the first (Test) and second (re-Test) administration. To maintain anonymity while also being able to match the questionnaires from the Test with the re-Test ones, a coding system was used. Participants were asked to provide the first letter of their name and the town they lived in and four digits of their phone number.

### 2.3. Instrument

The Sexual Violence Questionnaire in Sport has previously been validated and used in other studies [21,24,25] seeking to gain the study participants’ perceptions and experiences of sexual harassment in sport. In this study, we used the Spanish translation of the questionnaire [25], although a translation into Basque was also carried out. Similarly to this study, the current study used a sample of students studying for a sport-related degree to ensure the comprehensibility and appropriateness of the items. The pilot test confirmed the need to reformulate some terms in the Basque language in order to improve comprehension (e.g., ‘pinches’ or ‘diminutives’).

The questionnaire was divided into three sections. In the first section, perceptions of sexual harassment were collected in relation to 24 items about the coach behaviours. The unit of measurement for the perceptions was a scale: 0 = ‘I’m not sure if the behaviour is sexual violence’, 1 = ‘the behaviour doesn’t constitute sexual violence at all’, 2 = ‘maybe it is’, 3 = ‘the behaviour is almost certainly constitutes sexual violence’, and 4 = ‘the behaviour is almost certainly sexual violence’. The second section collected respondents’ experiences of the same 24 behaviours. Again, a different scale was also used: 0 = ‘I’m not sure’, 1 = ‘it has never happened to me’, 2 = ‘yes, sometimes’, 3 = ‘yes, often’ and 4 = ‘not to me, but to a colleague, yes’. ‘I’m not sure if the behaviour is sexual violence’ in perceptions and ‘I’m not sure’ in experiences were excluded for statistics due to their being added to the response options of the initially developed questionnaire. Finally, the third section was added to collect both independent variables and demographic data such as age, university degree, sport practiced, number of hours of sport per week, and level of competition, among others. 

### 2.4. Statistical Analysis

The results are presented as frequencies and percentages. The Kolmorov–Smirnov test was used to verify the normality of the data. Cronbach’s alpha test was used to analyse internal consistency, while the sign test (P) was used to determine whether there was a significant difference between the test and the re-test in each item of the perceptions/experiences. On the other hand, Cohen’s kappa analysis was used to analyse the concordance of the responses in each item. In this study, the interpretation of the scores is based on the classification system developed by [26]: scores of 0.09 indicate very poor agreement, scores of 0.01–0.20 indicate poor agreement, scores of 0.21–0.40 indicate fair agreement, values of 0.41–0.60 moderate agreement, scores 0.61–0.80 indicated substantial agreement, and scores 0.81–1.00 indicate almost perfect agreement. The analysis was performed using the Statistical Package for Social Sciences (SPSS Inc., version 26.0, Chicago, IL, USA). Significance was set at *p* < 0.05.

## 3. Results

### 3.1. Internal Consistency of the Sexual Violence Questionnaire in Sport

The internal consistency of the Sexual Violence Questionnaire in Sport was α = 0.891 for perceptions and α = 0.813 for experiences for all participants (see Table 1). In the case of females, the value of Cronbach’s alpha value of perceptions was slightly lower than that of males (α = 0.872 vs. 0.891, for females and males, respectively), while the value of experiences was slightly higher (α = 0.839 vs. 0.791, for females and males, respectively). It can be noted that the internal consistency of the questionnaire was good in all cases. 

### 3.2. Perceptions and Experiences of All Participants

For perceptions, significant differences between the Test and re-Test were found for eight of the 24 items (see Table 2). Similarly, in the case of experiences, significant differences between the Test and re-Test were in two of the 24 items (see Table 3). However, the Kappa index shows that agreement was significant for all items in the perceptions (range = 0.16–0.57, from mild to moderate agreement; *p* < 0.05). In the analysis, all items showed significant agreements with slightly higher values than in the perceptions (range = 0.37–0.61, from fair to substantial agreement; *p* < 0.05).

### 3.3. Perceptions and Experiences in Female Athletes

In contrast to the results obtained for the whole sample, in the case of females, significant differences between the Test and re-Test were identified in only 1 of the 24 items in relation to perceptions (see Table 4). In the case of experiences, significant differences between the Test and re-Test were found only two items (see Table 5). For both perceptions and experiences, Cohen’s Kappa indicated that agreement was significant for all items, except for item 24. The rest of the agreement values range from slight to substantial (range = 0.06–0.67; *p* < 0.05) for perceptions and from fair to substantial (range = 0.28–0.79; *p* < 0.05) for experiences.

### 3.4. Perceptions and Experiences in Male Athletes

For males, significant differences were found in 6 out of the 24 items, a higher number than compared to females (see Table 6). However, as with females, there were only two significant difference between the Test and the re-Test (see Table 7). For both perceptions and experiences, Cohen’s Kappa indicated that agreement was significant for all items, although the values of agreement ranged from slight to substantial (range = 0.18–0.65; *p* < 0.05) for perceptions and from fair to substantial (range = 0.33–0.65; *p* < 0.05) for experiences.

## 4. Discussion

This study evaluated the reliability of the Sexual Violence Questionnaire in Sport among Spanish-speaking athletes. Assessing the reliability of this questionnaire is essential to ensure its applicability in other cultural contexts, specifically in the Spanish context. The reliability of the Sexual Violence Questionnaire in Sport has been previously investigated (e.g., [19,26,27]), with internal consistency values exceeding 0.65 and below 0.90. These results align with those obtained in the present study, where the internal consistency of the questionnaire in Spanish and Basque was slightly higher in both languages (α > 0.791 for both perceptions and experiences in women and men). With regard to the reproducibility of the questionnaire, although some significant differences were observed between the Test and the re-Test (8 items in perceptions and 2 items in experiences), a significant agreement was observed, with a slight to moderate agreement in perceptions and a slightly higher level of from agreement in experiences, ranging from fair to substantial.

Sexual violence in the context of sports can be perpetrated or experienced by both males and females, regardless of the type of sport [3]. Nevertheless, it can be hypothesized that the perceptions and experiences of men and women are different, which may affect the reliability of the questionnaire. Regarding the internal consistency of the questionnaire, the results were higher than α > 0.791 for both perceptions and experiences in the whole sample, as well as for the female and male gender, indicating good values of internal consistency. Furthermore, the questionnaire can be reliable for both women and men, with very small differences between the two genders (Perceptions: α = 0.872 vs. 0.891; Experiences: α = 0.839 vs. 0.791, for women and men respectively). These values were higher than those previously obtained in the first validation study α > 0.65 [16] and similar to those obtained in another study [25], which recorded values between α = 0.77 and 0.89 for the different dimensions of the questionnaire in the perceptions section. These values show similarities with the reference literature in the Spanish context; however, further analysis is required to ascertain the suitability of the instrument in different or novel contexts.

In accordance with established guidelines [26], which indicate that the Test–re-Test is one of the most rigorous methods for measuring the reliability of an instrument over time, a comparative and concordance analysis was selected. In the analysis with all the participants, significant differences were observed between the Test and the re-Test for 8 of the 24 items. Conversely, only 2 items demonstrated significant differences in experiences. This trend was also observed for the female gender (differences in 1 item for perceptions and 2 items for experiences) and for the male gender (differences in 6 items for perceptions and 2 items for experiences) where more differences were observed for perceptions than for experiences. The analysis shows that the instrument is less reliable for perceptions. It is difficult to compare the data obtained because no studies were found that carried out a reproducibility analysis. However, in this study, the differences observed, especially in perceptions, may be due to obtaining fewer than five data points in an area, which could affect the results [27]. Additionally, perceptions do not have as much stability over time as experiences can demonstrate. This is because, as indicated by different authors, [28] perceptions of behaviours related to sexual violence in sports may change over time depending on the different sporting situations and levels of sport and how they change; even athletes may be more accepting of violence in more serious sporting situations, while experiences remain constant over time. Nevertheless, the instrument has been shown to be reliable for experiments, although further studies are needed to analyse the stability of the instrument over time. 

The discrepancies between the results of the reproducibility analysis and those of the reliability analysis are noteworthy. The Kappa index indicated substantial agreement about perceptions and experiences in the former. However, in the latter, the range of agreement was from slight to substantial. In the case of female respondents, no significant agreement was observed about item 24, either in perceptions or experiences. This item proposes sexual relations in exchange for privileges. As previously stated, the paucity of data in certain responses may have influenced the outcomes. However, the agreement was significant for the rest of the items, with slight and substantial agreement in *perceptions* and *experiences*. As with the female gender and in the total sample, the reliability of the questionnaire showed significant agreement for all items, with values ranging from mild to substantial for both *perceptions* and *experiences*. Despite the differences in the comparative analysis of the Test–re-Test, significant concordance relationships were also observed, albeit with low values in most cases. Therefore, further research is needed in this area.

Reliability studies typically assume that if the samples used have an equal number of men and women, the overall result when comparing the instruments could be valid for both genders. Consequently, several questionnaires have been validated with an exclusively female sample (e.g., [29]). However, this approach is somewhat questionable, as there are very few studies that have attempted to evaluate the reliability of questionnaires for both genders [30]. In this study, the internal consistency values for men are comparable to those for women with α = 0.891 for *perceptions* and relatively lower for experiences, with a value of α = 0.791. As was the case with the females, significant differences were observed in 8 items (7 in perceptions and 1 in experiences). With regard to the reliability of the instrument, the kappa value shows significant agreement for all items, with values ranging from *mild to substantial* (0.18–0.65; *p* < 0.05) for perceptions *and* from *fair to substantial* (0.33–0.65; *p* < 0.05) for experiences. These values determine the reliability and temporal stability of the instrument, although they are relatively lower than those obtained with the instrument validated to measure the harassment in team sports [31]. These discrepancies may be attributed to the fact that the original instrument was validated for the specific context of team sports, where group dynamics can influence perceptions. In contrast, this study encompasses a more diverse population in terms of individual and team sports, which could impact the consistency of responses due to varying experiences and contexts [32].

This study is not without its limitations: (1) the number of participants in the study; (2) the distribution of the sample according to gender (92 vs. 54); (3) the study’s focus on a specific sample from the University of the Basque Country, making it a case study (due to this, the findings may not be generalizable to other contexts); (4) the questionnaire being given in person (although the Sexual Violence Questionnaire in Sport was anonymized, it is possible that some participants gave socially desirable or biased answers, and this could have influenced the results, especially in terms of *perceptions*).

## 5. Conclusions

This study assessed the reliability of the Sexual Violence in Sport Questionnaire, showing a solid internal consistency across the sample and in gender segmentation. A gender-segmented analysis provides a deeper understanding of perceptions and experiences of sexual harassment in sport. While approximately half of the perceptions items do not appear to be stable over time, the experiences are stable over time, both in the sample as a whole and when differentiated according to gender. The Cohen’s Kappa analysis suggests concordance between responses at different times, which reinforces the validity of the Sexual Violence Questionnaire in Sport as a measurement tool. Despite these limitations, the results support the usefulness and reliability of the Sexual Violence Questionnaire in Sport and highlight the need for further research to analyse the reliability of the instrument on the basis of different population types. Based on the study’s findings, the practical implications suggest that the Sexual Violence Questionnaire in Sport, which has shown reliability within the Spanish-speaking athlete population, can be effectively used in similar contexts for assessing perceptions and experiences of sexual harassment. Sport organizations and educational institutions in Spanish-speaking regions can implement this tool to identify better and address issues of sexual violence, enabling targeted interventions and the development of preventive strategies tailored to cultural contexts. Additionally, this tool can aid in training coaches and athletes about appropriate behaviours, thereby fostering safer sports environments.

To improve the generalizability of these findings, future studies should consider including more diversified and representative samples in terms of gender, competitive level, and cultural context. This will allow for a broader and more accurate understanding of perceptions and experiences of sexual violence in sports across different contexts.

## Figures and Tables

**Table 1 ijerph-21-01214-t001:** Cronbach’s alpha values according to gender.

	Total		Women		Men	
Item	Test Percep	Test Experi	Test Percep	Test Experi	Test Percep	Test Experi
1	0.891	0.808	0.860	0.832	0893	0.788
2	0.891	0.808	0.861	0.832	0.893	0.788
3	0.886	0.809	0.867	0.834	0.883	0.791
4	0.895	0.813	0.862	0.842	0.897	0.789
5	0.889	0.811	0.867	0.825	0.890	0.799
6	0.885	0.810	0.864	0.848	0.885	0.779
7	0.881	0.802	0.861	0.831	0.880	0.777
8	0.882	0.799	0.865	0.834	0.879	0.767
9	0.880	0.807	0.867	0.831	0.877	0.789
10	0.891	0.804	0.866	0.837	0.891	0.776
11	0.889	0.803	0.866	0.839	0.888	0.774
12	0.890	0.802	0.866	0.839	0.889	0.770
13	0.889	0.803	0.860	0.837	0.890	0.776
14	0.884	0.805	0.868	0.830	0.882	0.787
15	0.883	0.809	0.862	0.839	0.883	0.787
16	0.886	0.811	0.876	0.837	0.883	0.786
17	0.885	0.806	0.868	0.827	0.883	0.790
18	0.880	0.801	0.867	0.824	0.878	0.784
19	0.881	0.800	0.873	0.825	0.878	0.780
20	0.885	0.799	0.879	0.828	0.881	0.773
21	0.882	0.805	0.869	0.830	0.879	0.784
22	0.883	0.810	0.872	0.836	0.881	0.789
23	0.884	0.806	0.870	0.829	0.882	0.789
24	0.885	0.810	0.874	0.834	0.883	0.789
Total	0.890	0.813	0.872	0.839	0.889	0.791

Note: Percep = perceptions of sexual harassment; Experi = sexual harassment experiences.

**Table 2 ijerph-21-01214-t002:** Perceptions of whole sample in the Test–Re-test.

	Not at All	Possibility	Constitutes	With Complete Certainty	I’m Not Sure	Sign Test (P)	Kappa
Touches athlete’s shoulder while instructing	78 (54.5%)	60 (42.0%)	3 (2.8%)	1 (0.7%)	3 (2.1%)	0.728	0.56 *
	81 (55.9%)	60 (41.4%)	3 (2.1%)	1 (0.7%)	1 (0.7%)		
Touches athlete’s shoulder when waving	97 (67.8%)	41 (28.7%)	4 (2.8%)	1 (0.7%)	3 (2.0%)	0.361	0.55 *
	97 (66.9%)	43 (29.7%)	4 (2.8%)	1 (0.7%)	1 (0.7%)		
Kisses athlete on the cheek	8 (5.6%)	62 (43.7%)	46 (32.4%)	26 (18.3%)	4 (2.8%)	0.222	0.28 *
	6 (4.2%)	72 (50.0%)	46 (31.9%)	20 (13.9%)	2 (1.4%)		
Hugs athlete when they win	93 (65.5%)	44 (31.0%)	3 (2.1%)	2 (1.4%)	4 (2.8%)	0.028 *	0.28 *
	78 (53.5%)	55 (38.5%)	6 (4.2%)	4 (2.8%)	3 (2.0%)		
Comes very close when instructing	26 (18.2%)	98 (68.5%)	16 (11.2%)	3 (2.1%)	3 (2.0%)	0.429	0.43 *
	30 (20.5%)	96 (67.6%)	13 (9.2%)	3 (2.1%)	4 (2.7%)		
Invites the athlete to a coffee	46 (32.4%)	79 (55.6%)	15 (10.6%)	2 (1.4%)	4 (2.7%)	0.791	0.30 *
	43 (29.9%)	84 (58.3%)	12 (8.3%)	5 (3.5%)	2 (1.4%)		
Invites the athlete to eat	26 (18.4%)	91 (64.5%)	21 (14.9%)	3 (2.1%)	5 (3.4%)	0.002 *	0.31 *
	22 (15.2%)	86 (58.6%)	26 (17.9%)	12 (8.3%)	1 (0.7%)		
Invites the athlete to dinner	22 (15.5%)	85 (59.9%)	30 (21.1%)	5 (3.5%)	4 (2.7%)	0.002 *	0.16 *
	13 (9.0%)	83 (57.2%)	32 (22.1%)	17 (11.7%)	1 (0.7%)		
Invites the athlete to the coach’s home	9 (6.3%)	65 (45.5%)	49 (34.3%)	20 (14.0%)	3 (2.0%)	0.596	0.41 *
	11 (7.5%)	64 (43.8%)	46 (31.5%)	25 (17.1%)	0 (0.0%)		
Asks the athlete about their leisure time	78 (53.5%)	55 (38.5%)	9 (6.3%)	1 (0.7%)	3 (2.0%)	0.022 *	0.40 *
	61 (43.0%)	69 (48.6%)	10 (7.0%)	2 (1.4%)	4 (2.7%)		
Asks the athlete about the weekend	86 (60.1%)	49 (34.3%)	8 (5.6%)	0 (0.0%)	3 (2.0%)	0.074	0.43 *
	74 (51.4%)	61 (42.4%)	6 (4.2%)	3 (2.1%)	2 (1.4%)		
Explains the coach’s personal weekend plans	69 (50.0%)	59 (42.8%)	10 (7.2%)	0 (0.0%)	8 (5.5%)	0.010 *	0.27 *
	60 (41.7%)	63 (43.8%)	19 (13.2%)	2 (1.4%)	2 (1.4%)		
Explains what the coach likes to do in their leisure time	89 (61.8%)	45 (31.3%)	8 (5.6%)	2 (1.4%)	2 (1.4%)	0.009 *	0.28 *
	68 (47.9%)	57 (40.1%)	13 (9.2%)	4 (2.8%)	4 (2.7%)		
Compliments the athlete’s physical appearance	14 (9.9%)	68 (47.9%)	49 (34.5%)	11 (7.7%)	4 (2.7%)	0.550	0.24 *
	15 (10.3%)	71 (49.0%)	42 (29.0%)	17 (11.7%)	1 (0.7%)		
Speaks with diminutives to the athlete	21 (15.3%)	80 (58.4%)	26 (19.0%)	10 (7.3%)	9 (6.1%)	0.596	0.31 *
	20 (13.9%)	79 (54.9%)	33 (22.9%)	12 (8.3%)	2 (1.4%)		
Makes derogatory comments about women	8 (5.7%)	16 (11.3%)	47 (33.3%)	70 (49.6%)	5 (3.4%)	0.061	0.30 *
	8 (5.6%)	23 (16.0%)	60 (41.7%)	53 (36.8%)	2 (1.4%)		
Pinches the athlete	13 (9.4%)	56 (40.3%)	52 (37.4%)	18 (12.9%)	7 (4.8%)	0.349	0.21 *
	15 (10.5%)	61 (42.7%)	52 (36.4%)	15 (10.5%)	3 (2.0%)		
Massages the athlete’s back	16 (11.4%)	59 (42.1%)	43 (30.7%)	22 (15.7%)	6 (4.1%)	0.382	0.31 *
	12 (8.4%)	76 (53.1%)	38 (26.6%)	17 (11.9%)	3 (2.0%)		
Asks the athlete questions about their sex life	6 (4.1%)	23 (15.9%)	52 (35.9%)	64 (44.1%)	1 (0.7%)	1.000	0.29 *
	8 (5.5%)	19 (13.0%)	57 (39.0%)	62 (42.5%)	1 (0.7%)		
Stare at the athlete’s breasts or backside	6 (4.1%)	11 (7.5%)	36 (24.7%)	93 (63.7%)	0 (0.0%)	1.000	0.24 *
	7 (4.8%)	7 (4.8%)	42 (29.0%)	89 (61.4%)	1 (0.7%)		
Shows sexual interest in the athlete	5 (3.4%)	8 (5.5%)	22 (15.1%)	111 (76.0%)	0 (0.0%)	0.194	0.27 *
	7 (4.8%)	5 (3.4%)	35 (24.0%)	99 (67.8%)	0 (0.0%)		
Kisses the athlete on the lips	5 (3.4%)	3 (2.1%)	16 (11.0%)	122 (83.6%)	0 (0.0%)	0.006 *	0.45 *
	5 (3.4%)	6 (4.1%)	29 (19.9%)	106 (72.6%)	0 (0.0%)		
Proposes sex with nothing in return	5 (3.4%)	6 (4.1%)	16 (11.0%)	119 (81.5%)	0 (0.0%)	0.188	0.34 *
	6 (4.1%)	2 (1.4%)	31 (21.2%)	107 (73.3%)	0 (0.0%)		
Proposes sexual relations in exchange for privileges	5 (3.4%)	1 (0.7%)	5 (3.4%)	135 (92.5%)	0 (0.0%)	0.007 *	0.30 *
	6 (4.1%)	0 (0.0%)	19 (13.0%)	121 (82.9%)	0 (0.0%)		

Note: * *p* < 0.05 significant differences between the Test and the re-Test; Not at all: The behaviour does not constitute sexual harassment at all; Possibility: The behaviour may constitute sexual harassment; It constitutes: The behaviour constitutes sexual harassment; With complete certainty: You believe with complete certainty that the behaviour constitutes sexual harassment.

**Table 3 ijerph-21-01214-t003:** Experiences of whole sample in the Test–Re-test.

	It’s Never Happened to Me	Ever	Often	Not Me, Others Do	I’m Not Sure	Sign Test (P)	Kappa
Touches the athlete’s shoulder while instructing	14 (9.6%)	77 (52.7%)	53 (36.3%)	2 (1.4%)	0 (0.0%)	0.203	0.40 *
	9 (6.2%)	78 (53.4%)	58 (39.7%)	1 (0.7%)	0 (0.0%)		
Touches the athlete’s shoulder when waving	9 (6.2%)	65 (44.5%)	69 (47.3%)	3 (2.1%)	3 (2.0%)	0.212	0.37 *
	5 (3.4%)	84 (57.5%)	56 (38.4%)	1 (0.7%)	0 (0.0%)		
Kisses the athlete on the cheek	115 (79.3%)	26 (17.9%)	1 (0.7%)	3 (2.1%)	1 (0.7%)	0.078	0.61 *
	107 (73.8%)	34 (23.4%)	1 (0.7%)	3 (2.1%)	1 (0.7%)		
Hugs the athlete when they win	10 (6.9%)	47 (32.4%)	86 (59.3%)	2 (1.4%)	1 (0.7%)	0.349	0.48 *
	11 (7.6%)	51 (35.2%)	82 (56.6%)	1 (0.7%)	1 (0.7%)		
Comes very close when instructing	45 (32.4%)	68 (48.9%)	23 (16.5%)	3 (2.2%)	7 (4.8%)	0.671	0.43 *
	44 (30.6%)	70 (48.6%)	27 (18.8%)	3 (2.1%)	2 (1.4%)		
Invites the athlete to a coffee	45 (31.3%)	79 (54.9%)	16 (11.1%)	4 (2.8%)	2 (1.4%)	0.643	0.51 *
	47 (32.2%)	79 (54.1%)	16 (11.0%)	4 (2.7%)	0 (0.0%)		
Invites the athlete to eat	93 (64.6%)	38 (26.4%)	9 (6.3%)	4 (2.8%)	2 (1.4%)	0.864	0.53 *
	94 (65.3%)	40 (27.8%)	6 (4.2%)	4 (2.8%)	2 (1.4%)		
Invites the athlete to dinner	99 (68.8%)	34 (23.6%)	7 (4.9%)	4 (2.8%)	2 (1.4%)	1.000	0.50 *
	101 (70.1%)	32 (22.2%)	6 (4.2%)	5 (3.5%)	2 (1.4%)		
Invites the athlete to the coach’s home	122 (84.7%)	16 (11.1%)	3 (2.1%)	3 (2.1%)	2 (1.4%)	0.093	0.48 *
	116 (80.0%)	23 (15.9%)	3 (2.1%)	3 (2.1%)	1 (0.7%)		
Asks the athlete about their leisure time	13 (9.2%)	73 (51.4%)	53 (37.3%)	3 (2.1%)	5 (3.4%)	0.212	0.52 *
	20 (13.7%)	73 (50.0%)	51 (34.9%)	2 (1.4%)	0 (0.0%)		
Asks the athlete about their weekend	12 (8.5%)	70 (49.6%)	57 (40.4%)	2 (1.4%)	6 (4.1%)	0.049 *	0.55 *
	14 (9.7%)	80 (55.6%)	48 (33.3%)	2 (1.4%)	2 (1.4%)		
Explains the coach’s personal weekend plans	47 (33.1%)	71 (50.0%)	24 (16.9%)	0 (0.0%)	4 (2.7%)	0.030 *	0.45 *
	37 (25.7%)	77 (53.5%)	30 (20.8%)	0 (0.0%)	2 (1.4%)		
Explains what the coach likes to do in their leisure time	37 (25.7%)	75 (52.1%)	31 (21.5%)	1 (0.7%)	2 (1.4%)	1.000	0.35 *
	35 (24.3%)	80 (55.6%)	29 (20.1%)	0 (0.0%)	2 (1.4%)		
Compliments the athlete’s physical appearance	78 (56.1%)	49 (35.3%)	6 (4.3%)	6 (4.3%)	8 (5.4%)	0.871	0.51 *
	73 (51.4%)	60 (42.3%)	2 (1.4%)	7 (4.9%)	5 (3.4%)		
Speaks with diminutives to the athlete	87 (62.6%)	40 (28.8%)	4 (2.9%)	8 (5.8%)	7 (4.8%)	0.874	0.44 *
	93 (65.0%)	37 (25.9%)	6 (4.2%)	7 (4.9%)	4 (2.7%)		
Makes derogatory comments about women	106 (74.1%)	30 (21.0%)	3 (2.1%)	4 (2.8%)	3 (2%)	0.845	0.55 *
	104 (74.3%)	28 (20.0%)	3 (2.1%)	5 (3.6%)	6 (4.1%)		
Pinches the athlete	118 (83.1%)	17 (12.0%)	1 (0.7%)	6 (4.2%)	4 (2.7%)	1.000	0.46 *
	117 (82.4%)	19 (13.4%)	1 (0.7%)	5 (3.5%)	4 (2.7%)		
Massages the athlete’s back	98 (68.5%)	37 (25.9%)	1 (0.7%)	7 (4.9%)	3 (2%)	0.855	0.54 *
	98 (68.1%)	39 (27.1%)	2 (1.4%)	5 (3.5%)	3 (2.0%)		
Asks the athlete questions about their sex life	114 (78.6%)	21 (14.5%)	4 (2.8%)	6 (4.1%)	1 (0.7%)	1.000	0.51 *
	111 (78.2%)	27 (19.0%)	4 (2.8%)	0 (0.0%)	4 (2.7%)		
Stares at the athlete’s breasts or backside	112 (81.2%)	14 (10.1%)	5 (3.6%)	7 (5.1%)	8 (5.4%)	0.523	0.47 *
	118 (83.7%)	12 (8.5%)	5 (3.5%)	6 (4.3%)	5 (3.4%)		
Shows sexual interest in the athlete	132 (91.7%)	4 (2.8%)	2 (1.4%)	6 (4.2%)	2 (1.4%)	1.000	0.38 *
	135 (92.5%)	4 (2.7%)	3 (2.1%)	4 (2.7%)	0 (0.0%)		
Kisses the athlete on the lips	141 (96.6%)	1 (0.7%)	0 (0.0%)	4 (2.7%)	0 (0.0%)	1.000	0.69 *
	140 (96.6%)	2 (1.4%)	1 (0.7%)	2 (1.4%)	1 (0.7%)		
Proposes sex with nothing in return	140 (95.9%)	1 (0.7%)	0 (0.0%)	5 (3.4%)	0 (0.0%)	1.000	0.57 *
	140 (95.9%)	3 (2.1%)	1 (0.7%)	2 (1.4%)	0 (0.0%)		
Proposes sexual relations in exchange for privileges	142 (97.3%)	0 (0.0%)	0 (0.0%)	4 (2.7%)	0 (0.0%)	0.625	0.42 *
	143 (97.9%)	1 (0.7%)	1 (0.7%)	1 (0.7%)	0 (0.0%)		

Note: * *p* < 0.05 significant differences between the Test and the re-Test. Sometimes: Yes. Ever. Often: Yes. Often.

**Table 4 ijerph-21-01214-t004:** Female Perceptions in the Test–Re-test.

	Not at All	Possibility	Constitutes	With Complete Certainty	I’m Not Sure	Sign Test (P)	Kappa
Touches the athlete’s shoulder while instructing	24 (49.0%)	24 (49.0%)	0 (0.0%)	1 (2.0%)	3 (5.9%)	1.000	0.41 *
	27(52.9%)	22 (43.1%)	1 (2.0%)	1 (2.0%)	1 (1.9%)		
Touches the athlete’s shoulder when waving	32 (65.3%)	16 (32.7%)	1 (2.0%)	0 (0.0%)	3 (5.8%)	0.508	0.61 *
	33 (64.7%)	15 (29.4%)	2 (3.9%)	1 (2.0%)	1 (1.9%)		
Kisses the athlete on the cheek	2 (4.1%)	23 (46.9%)	17 (34.7%)	7 (14.3%)	3 (5.9%)	0.664	0.30 *
	1 (2.0%)	27 (54.0%)	15 (30.0%)	7 (14.0%)	2 (3.8%)		
Hugs the athlete when they win	36 (72.0%)	13 (26.0%)	0 (0.0%)	1 (2.0%)	2 (3.8%)	0.143	0.25 *
	29 (59.2%)	16 (32.7%)	1 (2.0%)	3 (6.1%)	3 (5.8%)		
Comes very close when instructing	5 (9.8%)	36 (70.6%)	7 (13.7%)	3 (5.9%)	1 (1.9%)	0.065	0.51 *
	8 (16.3%)	35 (71.4%)	4 (8.2%)	2 (4.1%)	3 (5.8%)		
Invites the athlete to a coffee	13 (26.5%)	30 (61.2%)	5 (10.2%)	1 (2.0%)	3 (5.9%)	1.000	0.46 *
	15 (29.4%)	28 (54.9%)	4 (7.8%)	4 (7.8%)	1 (1.9%)		
Invites the athlete to eat	4 (8.3%)	35 (72.9%)	8 (16.7%)	1 (2.1%)	4 (7.7%)	0.096	0.27 *
	5 (9.6%)	33 (63.5%)	7 (13.5%)	7 (13.5%)	0 (0.0%)		
Invites the athlete to dinner	4 (8.3%)	31 (64.6%)	11 (22.9%)	2 (4.2%)	4 (7.8%)	0.078	0.67 *
	3 (5.8%)	30 (57.7%)	10 (19.2%)	9 (17.3%)	0 (0.0%)		
Invites the athlete to the coach’s home	1 (2.0%)	19 (38.0%)	22 (44.0%)	8 (16.0%)	2 (3.8%)	0.503	0.39 *
	1 (1.9%)	20 (38.5%)	20 (38.5%)	11 (21.2%)	0 (0.0%)		
Asks the athlete about their leisure time	27 (54.0%)	19 (38.0%)	3 (6.0%)	1 (2.0%)	2 (3.8%)	1.000	0.40 *
	26 (52.0%)	21 (42.0%)	2 (4.0%)	1 (2.0%)	2 (3.8%)		
Asks the athlete about their weekend	32 (65.3%)	15 (30.6%)	2 (4.1%)	0 (0.0%)	3 (5.8%)	0.424	0.43 *
	30 (58.8%)	19 (37.3%)	0 (0.0%)	2 (3.9%)	1 (1.9%)		
Explains the coach’s personal weekend plans	25 (53.2%)	19 (40.4%)	3 (6.4%)	0 (0.0%)	5 (9.6%)	0.167	0.29 *
	22 (44.0%)	22 (44.0%)	4 (8.0%)	2 (4.0%)	2 (3.8%)		
Explains what the coach likes to do in their leisure time	34 (66.7%)	14 (27.5%)	2(3.9%)	1 (2.0%)	1 (1.9%)	0.481	0.26 *
	30 (60.0%)	17 (34.0%)	1 (2.0%)	2 (4.0%)	2 (3.8%)		
Compliments the athlete’s physical appearance	3 (5.9%)	22 (43.1%)	23 (45.1%)	3 (5.9%)	1 (1.9%)	0.064	0.27 *
	4 (7.8%)	16 (31.4%)	22 (43.1%)	9 (17.6%)	1 (1.9%)		
Speaks with diminutives to the athlete	3 (6.3%)	34 (70.8%)	9 (18.8%)	2 (4.2%)	4 (7.7%)	0.648	0.21 *
	4 (7.8%)	31 (60.8%)	10 (19.6%)	6 (11.8%)	1 (1.9%)		
Makes derogatory comments about women	1 (2.0%)	1 (2.0%)	21 (42.9%)	26 (53.1%)	3 (5.8%)	0.629	0.37 *
	1 (2.0%)	5 (9.8%)	19 (37.3%)	26 (51.0%)	1 (1.9%)		
Pinches the athlete	2 (4.3%)	20 (42.6%)	19 (40.4%)	6 (12.8%)	5 (9.8%)	0.405	0.22 *
	5 (10.0%)	19 (38.0%)	21 (42.0%)	5 (10.0%)	2 (3.8%)		
Massages the athlete’s back	3 (6.0%)	18 (36.0%)	19 (38.0%)	10 (20.0%)	2 (3.8%)	0.108	0.26 *
	4 (8.0%)	29 (58.0%)	7 (14.0%)	10 (20.0%)	2 (3.8%)		
Asks the athlete questions about their sex life	1 (1.9%)	2 (3.8%)	20 (38.5%)	29 (55.8%)	0 (0.0%)	0.664	0.28 *
	2 (3.8%)	3 (5.8%)	20 (38.5%)	27 (51.9%)	0 (0.0%)		
Stares at the athlete’s breasts or backside	0 (0.0%)	3 (5.8%)	10 (19.2%)	39 (75.0%)	0 (0.0%)	1.000	0.06 *
	1 (1.9%)	2 (3.8%)	11 (21.2%)	38 (73.1%)	0 (0.0%)		
Shows sexual interest in the athlete	0 (0.0%)	1 (1.9%)	8 (15.4%)	43 (82.7%)	0 (0.0%)	0.791	0.13 *
	1 (1.9%)	2 (3.8%)	7 (13.5%)	42 (80.8%)	0 (0.0%)		
Kisses the athlete on the lips	0 (0.0%)	1 (1.9%)	3 (5.8%)	48 (92.3%)	0 (0.0%)	0.219	0.45 *
	1 (1.9%)	0 (0.0%)	7 (13.5%)	44 (84.6%)	0 (0.0%)		
Proposes sex with nothing in return	0 (0.0%)	0 (0.0%)	8 (15.4%)	44 (84.6%)	0 (0.0%)	1.000	0.31 *
	1 (1.9%)	0 (0.0%)	6 (11.5%)	45 (86.5%)	0 (0.0%)		
Proposes sexual relations in exchange for privileges	0 (0.0%)	0 (0.0%)	0 (0.0%)	52 (100%)	0 (0.0%)	0.031 *	0.00
	1 (1.9%)	0 (0.0%)	5 (9.6%)	46 (88.5%)	0 (0.0%)		

Note: * *p* < 0.05 significant differences between the Test and the re-Test. Not at all: the behaviour does not constitute sexual harassment at all. Possibility: the behaviour may constitute sexual harassment. It constitutes: the behaviour constitutes sexual harassment. With complete certainty: you believe with complete certainty that the behaviour constitutes sexual harassment.

**Table 5 ijerph-21-01214-t005:** Female experiences in the Test–Re-test.

	It’s Never Happened to Me	Ever	Often	Not Me, Others Do	I’m Not Sure	Sign Test (P)	Kappa
Touches the athlete’s shoulder while instructing	2 (3.8%)	33 (63.5%)	16 (30.8%)	1 (1.9%)	0 (0.0%)	0.607	0.44 *
	2 (3.8%)	30 (57.7%)	20 (38.5%)	0 (0.0%)	0 (0.0%)		
Touches the athlete’s shoulder when waving	1 (1.9%)	27 (51.9%)	23 (44.2%)	1 (1.9%)	0 (0.0%)	0.143	0.37 *
	1 (1.9%)	34 (65.4%)	17 (32.7%)	0 (0.0%)	0 (0.0%)		
Kisses the athlete on the cheek	32 (61.5%)	19 (36.5%)	0 (0.0%)	1 (1.9%)	0 (0.0%)	1.000	0.68 *
	31 (59.6%)	20 (39.2%)	0 (0.0%)	0 (0.0%)	1 (1.9%)		
Hugs the athlete when they win	2 (3.9%)	18 (35.3%)	31 (60.8%)	0 (0.0%)	1 (1.9%)	1.000	0.46 *
	2 (3.9%)	18 (35.3%)	31 (60.8%)	0 (0.0%)	1 (1.9%)		
Comes very close when instructing	19 (41.3%)	17 (37.0%)	9 (19.6%)	1 (2.2%)	6 (11.5%)	0.774	0.60 *
	19 (38.0%)	20 (40.0%)	11 (22.0%)	0 (0.0%)	2 (3.8%)		
Invites the athlete to a coffee	17 (32.7%)	24 (46.2%)	9 (17.3%)	2 (3.8%)	0 (0.0%)	0.629	0.47 *
	14 (26.9%)	31 (59.6%)	5 (9.6%)	2 (3.8%)	0 (0.0%)		
Invites the athlete to eat	34 (65.4%)	13 (25.0%)	4 (7.7%)	1 (1.9%)	0 (0.0%)	1.000	0.46 *
	35 (68.6%)	13 (25.5%)	2 (3.9%)	1 (2.0%)	1 (1.9%)		
Invites the athlete to dinner	38 (73.1%)	11 (21.2%)	2 (3.8%)	1 (1.9%)	0 (0.0%)	1.000	0.42 *
	41 (80.4%)	7 (13.7%)	1 (2.0%)	2 (3.9%)	1 (2.0%)		
Invites the athlete to the coach’s home	43 (82.7%)	7 (13.5%)	1 (1.9%)	1 (1.9%)	0 (0.0%)	1.000	0.64 *
	45 (86.5%)	6 (11.5%)	0 (0.0%)	1 (1.9%)	0 (0.0%)		
Asks the athlete about their leisure time	3 (6.1%)	26 (53.1%)	19 (38.8%)	1 (2.0%)	3 (5.8%)	0.388	0.57 *
	5 (9.6%)	27 (51.9%)	19 (36.5%)	1 (1.9%)	0 (0.0%)		
Asks the athlete about their weekend	3 (6.3%)	23 (47.9%)	21 (43.8%)	1 (2.1%)	4 (7.7%)	0.344	0.64 *
	4 (7.8%)	27 (52.9%)	20 (39.2%)	0 (0.0%)	1 (1.9%)		
Explains the coach’s personal weekend plans	13 (26.5%)	26 (53.1%)	10 (20.4%)	0 (0.0%)	3 (5.8%)	0.021 *	0.47 *
	9 (18.0%)	24 (48.0%)	17 (34.0%)	0 (0.0%)	2 (3.9%)		
Explains what the coach likes to do in their leisure time	12 (24.0%)	23 (46.0%)	14 (28.0%)	1 (2.0%)	2 (3.8%)	0.263	0.36 *
	7 (14.0%)	27 (54.0%)	16 (32.0%)	0 (0.0%)	2 (3.8%)		
Compliments the athlete’s physical appearance	24 (50.0%)	18 (37.5%)	2 (4.2%)	4 (8.3%)	4 (7.7%)	0.581	0.53 *
	24 (49.0%)	21 (42.9%)	0 (0.0%)	4 (8.2%)	3 (5.8%)		
Speaks with diminutives to the athlete	25 (54.3%)	13 (28.3%)	3 (6.5%)	5 (10.9%)	6 (11.8%)	0.035 *	0.41 *
	34 (69.4%)	10 (20.4%)	2 (4.1%)	3 (6.1%)	3 (5.8%)		
Makes derogatory comments about women	40 (81.6%)	8 (16.3%)	0 (0.0%)	1 (2.0%)	3 (5.8%)	0.754	0.28 *
	41 (85.4%)	4 (8.3%)	0 (0.0%)	3 (6.3%)	4 (7.7%)		
Pinches the athlete	40 (80.0%)	6 (12.0%)	0 (0.0%)	4 (8.0%)	2 (3.8%)	1.000	0.60 *
	38 (77.6%)	8 (16.3%)	0 (0.0%)	3 (6.1%)	3 (5.8%)		
Massages the athlete’s back	28 (54.9%)	18 (35.3%)	1 (2.0%)	4 (7.8%)	1 (1.9%)	0.607	0.47 *
	31 (62.0%)	13 (26.0%)	2 (4.0%)	4 (8.0%)	2 (3.8%)		
Asks the athlete questions about their sex life	43 (82.7%)	4 (7.7%)	2 (3.8%)	3 (5.8%)	0 (0.0%)	0.727	0.41 *
	43 (87.8%)	4 (8.2%)	2 (4.1%)	0 (0.0%)	3 (5.8%)		
Stares at the athlete’s breasts or backside	40 (85.1%)	4 (8.5%)	2 (4.3%)	1 (2.1%)	5 (9.8%)	1.000	0.55 *
	40 (81.6%)	6 (12.2%)	2 (4.1%)	1 (2.0%)	3 (5.8%)		
Shows sexual interest in the athlete	46 (90.2%)	3 (5.9%)	0 (0.0%)	2 (3.9%)	1 (1.9%)	1.000	0.53 *
	47 (90.4%)	3 (5.8%)	1 (1.9%)	1 (1.9%)	0 (0.0%)		
Kisses the athlete on the lips	50 (96.2%)	1 (1.9%)	0 (0.0%)	1 (1.9%)	0 (0.0%)	1.000	0.79 *
	49 (94.2%)	2 (3.8%)	0 (0.0%)	1 (1.9%)	0 (0.0%)		
Proposes sex with nothing in return	49 (94.2%)	1 (1.9%)	0 (0.0%)	2 (3.8%)	0 (0.0%)	1.000	0.65 *
	49 (94.2%)	2 (3.8%)	0 (0.0%)	1 (1.9%)	0 (0.0%)		
Proposes sexual relations in exchange for privileges	51 (98.1%)	0 (0.0%)	0 (0.0%)	1 (1.9%)	0 (0.0%)	1.000	−0.01
	51 (98.1%)	1 (1.9%)	0 (0.0%)	0 (0.0%)	0 (0.0%)		

Note: * *p* < 0.05 significant differences between the Test and the re-Test. Sometimes: Yes. Ever. Often: Yes. Often.

**Table 6 ijerph-21-01214-t006:** Male perceptions in the Test–Re-test.

	Not at All	Possibility	Constitutes	With Complete Certainty	I’m Not Sure	Sign Test (P)	Kappa
Touches the athlete’s shoulder while instructing	54 (57.4%)	36 (38.3%)	4 (4.3%)	0 (0.0%)	0 (0.0%)	0.815	0.63 *
	54 (57.4%)	38 (40.4%)	2 (2.1%)	0 (0.0%)	0 (0.0%)		
Touches the athlete’s shoulder when waving	65 (69.1%)	25 (26.6%)	3 (3.2%)	1 (1.1%)	0 (0.0%)	0.664	0.51 *
	64 (68.1%)	28 (29.8%)	2 (2.1%)	0 (0.0%)	0 (0.0%)		
Kisses the athlete on the cheek	6 (6.5%)	39 (41.9%)	29 (31.2%)	19 (20.4%)	1 (1.1%)	0.302	0.27 *
	5 (5.3%)	45 (47.9%)	31 (33.0%)	13 (13.8%)	0 (0.0%)		
Hugs the athlete when they win	57 (62.0%)	31 (33.7%)	3 (3.3%)	1 (1.1%)	2 (2.1%)	0.134	0.28 *
	49 (52.1%)	39 (41.5%)	5 (5.3%)	1 (1.1%)	0 (0.0%)		
Comes very close when instructing	21 (22.8%)	62 (67.4%)	9 (9.8%)	0 (0.0%)	2 (2.1%)	1.000	0.37 *
	22 (23.7%)	61 (65.6%)	9 (9.7%)	1 (1.1%)	1 (1.1%)		
Invites the athlete to a coffee	33 (35.5%)	49 (52.7%)	10 (10.8%)	1 (1.1%)	1 (1.1%)	0.877	0.21 *
	28 (30.1%)	56 (60.2%)	8 (8.6%)	1 (1.1%)	1 (1.1%)		
Invites the athlete to eat	22 (23.7%)	56 (60.2%)	13 (14.0%)	2 (2.2%)	1 (1.1%)	0.015 *	0.32 *
	17 (18.3%)	52 (55.9%)	19 (20.4%)	5 (5.4%)	1 (1.1%)		
Invites the athlete to dinner	18 (19.1%)	54 (57.4%)	19 (20.2%)	3 (3.2%)	0 (0.0%)	0.017 *	0.20 *
	10 (10.8%)	53 (57.0%)	22 (23.7%)	8 (8.6%)	1 (1.1%)		
Invites the athlete to the coach’s home	8 (8.6%)	46 (49.5%)	27 (29.0%)	12 (12.9%)	1 (1.1%)	1.000	0.40 *
	10 (10.6%)	44 (46.8%)	26 (27.7%)	14 (14.9%)	0 (0.0%)		
Asks the athlete about their leisure time	51 (54.8%)	36 (38.7%)	6 (6.5%)	0 (0.0%)	1 (1.1%)	0.005 *	0.40 *
	35 (38.0%)	48 (52.2%)	8 (8.7%)	1 (1.1%)	2 (2.1%)		
Asks the athlete about their weekend	54 (57.4%)	34 (36.2%)	6 (6.4%)	0 (0.0%)	0 (0.0%)	0.151	0.42 *
	44 (47.3%)	42 (45.2%)	6 (6.5%)	1 (1.1%)	1 (1.1%)		
Explains the coach’s personal weekend plans	44 (48.4%)	40 (44.0%)	7 (7.7%)	0 (0.0%)	3 (3.2%)	0.045 *	0.26 *
	38 (40.4%)	41 (43.6%)	15 (16.0%)	0 (0.0%)	0 (0.0%)		
Explains what the coach likes to do in their leisure time	55 (59.1%)	31 (33.3%)	6 (6.5%)	1 (1.1%)	1 (1.1%)	0.012 *	0.27 *
	38 (41.3%)	40 (43.5%)	12 (13.0%)	2 (2.2%)	2 (2.1%)		
Compliments the athlete’s physical appearance	11 (12.1%)	46 (50.5%)	26 (28.6%)	8 (8.8%)	3 (3.2%)	0.658	0.20 *
	11 (11.7%)	55 (58.5%)	20 (21.3%)	8 (8.5%)	0 (0.0%)		
Speaks with diminutives to the athlete	18 (20.2%)	46 (51.7%)	17 (19.1%)	8 (9.0%)	5 (5.3%)	0.871	0.34 *
	16 (17.2%)	48 (51.6%)	23 (24.7%)	6 (6.5%)	1 (1.1%)		
Makes derogatory comments about women	7 (7.6%)	15 (16.3%)	26 (28.3%)	44 (47.8%)	2 (2.1%)	0.080	0.27 *
	7 (7.5%)	18 (19.4%)	41 (44.1%)	27 (29.0%)	1 (1.1%)		
Pinches the athlete	11 (12.0%)	36 (39.1%)	33 (35.9%)	12 (13.0%)	2 (2.1%)	0.671	0.20 *
	10 (10.8%)	42 (45.2%)	31 (33.3%)	10 (10.8%)	1 (1.1%)		
Massages the athlete’s back	13 (14.4%)	41 (45.6%)	24 (26.7%)	12 (13.3%)	4 (4.2%)	1.000	0.35 *
	8 (8.6%)	47 (50.5%)	31 (33.3%)	7 (7.5%)	1 (1.1%)		
Asks the athlete questions about their sex life	5 (5.4%)	21 (22.6%)	32 (34.4%)	35 (37.6%)	1 (1.1%)	0.883	0.29 *
	6 (6.4%)	16 (17.0%)	37 (39.4%)	35 (37.2%)	1 (1.1%)		
Stares at the athlete’s breasts or backside	6 (6.4%)	8 (8.5%)	26 (27.7%)	54 (57.4%)	0 (0.0%)	1.000	0.30 *
	6 (6.5%)	5 (5.4%)	31 (33.3%)	51 (54.6%)	1 (1.1%)		
Shows sexual interest in the athlete	5 (5.3%)	7 (7.4%)	14 (14.9%)	68 (72.3%)	0 (0.0%)	0.230	0.31 *
	6 (6.4%)	3 (3.2%)	28 (29.8%)	57 (60.6%)	0 (0.0%)		
Kisses the athlete on the lips	5 (5.3%)	2 (2.1%)	13 (13.8%)	74 (78.7%)	0 (0.0%)	0.023 *	0.44 *
	4 (4.3%)	6 (6.4%)	22 (23.4%)	62 (66.0%)	0 (0.0%)		
Proposes sex with nothing in return	5 (5.3%)	6 (6.4%)	8 (8.5%)	75 (79.8%)	0 (0.0%)	0.089	0.35 *
	5 (5.3%)	2 (2.1%)	25 (26.6%)	62 (66.0%)	0 (0.0%)		
Proposes sexual relations in exchange for privileges	5 (5.3%)	1 (1.1%)	5 (5.3%)	83 (88.3%)	0 (0.0%)	0.096	0.35 *
	5 (5.3%)	0 (0.0%)	14 (14.9%)	75 (79.8%)	0 (0.0%)		

Note: * *p* < 0.05 significant differences between the Test and the re-Test. Not at all: the behaviour does not constitute sexual harassment at all. Possibility: the behaviour may constitute sexual harassment. It constitutes: the behaviour constitutes sexual harassment. With complete certainty: you believe with complete certainty that the behaviour constitutes sexual harassment.

**Table 7 ijerph-21-01214-t007:** Male experiences in the Test–Re-test.

	It’s Never Happened to Me	Ever	Often	Not Me, Others Do	I’m Not Sure	Sign Test (P)	Kappa
Touches the athlete’s shoulder while instructing	12 (12.8%)	44 (46.8%)	37 (39.4%)	1 (1.1%)	0 (0.0%)	0.310	0.38 *
	7 (7.4%)	48 (51.1%)	38 (40.4%)	1 (1.1%)	0 (0.0%)		
Touches the athlete’s shoulder when waving	8 (8.5%)	38 (40.4%)	46 (48.9%)	2 (2.1%)	0 (0.0%)	0.735	0.36 *
	4 (4.3%)	50 (53.2%)	39 (41.5%)	1 (1.1%)	0 (0.0%)		
Kisses the athlete on the cheek	83 (89.2%)	7 (7.5%)	1 (1.1%)	2 (2.2%)	1 (1.1%)	0.022 *	0.48 *
	76 (80.9%)	14 (14.9%)	1 (1.1%)	3 (3.2%)	0 (0.0%)		
Hugs the athlete when they win	8 (8.5%)	29 (30.9%)	55 (58.5%)	2 (2.1%)	0 (0.0%)	0.248	0.50 *
	9 (9.6%)	33 (35.1%)	51 (54.3%)	1 (1.1%)	0 (0.0%)		
Comes very close when instructing	26 (28.0%)	51 (54.8%)	14 (15.1%)	2 (2.2%)	1 (1.1%)	0.871	0.33 *
	25 (26.6%)	50 (53.2%)	16 (17.0%)	3 (3.2%)	0 (0.0%)		
Invites the athlete to a coffee	28 (30.4%)	55 (59.8%)	7 (7.6%)	2 (2.2%)	2 (2.1%)	1.000	0.54 *
	33 (35.1%)	48 (51.1%)	11 (11.7%)	2 (2.1%)	0 (0.0%)		
Invites the athlete to eat	59 (64.1%)	25 (27.2%)	5 (5.4%)	3 (3.3%)	2 (2.1%)	1.000	0.56 *
	59 (63.4%)	27 (29.0%)	4 (4.3%)	3 (3.2%)	1 (1.1%)		
Invites the athlete to dinner	61 (66.3%)	23 (25.0%)	5 (5.4%)	3 (3.3%)	2 (2.1%)	0.832	0.53 *
	60 (64.5%)	25 (26.9%)	5 (5.4%)	3 (3.2%)	1 (1.1%)		
Invites the athlete to the coach’s home	79 (85.9%)	9 (9.8%)	2 (2.2%)	2 (2.2%)	2 (2.1%)	0.031 *	0.40 *
	71 (76.3%)	17 (18.3%)	3 (3.2%)	2 (2.2%)	1 (1.1%)		
Asks the athlete about their leisure time	10 (10.8%)	47 (50.5%)	34 (36.6%)	2 (2.2%)	2 (2.1%)	0.458	0.49 *
	15 (16.0%)	46 (48.9%)	32 (34.0%)	1 (1.1%)	0 (0.0%)		
Asks the athlete about their weekend	9 (9.7%)	47 (50.5%)	36 (38.7%)	1 (1.1%)	2 (2.1%)	0.124	0.50 *
	10 (10.8%)	53 (57.0%)	28 (30.1%)	2 (2.2%)	1 (1.1%)		
Explains the coach’s personal weekend plans	34 (36.6%)	45 (48.4%)	14 (15.1%)	0 (0.0%)	1 (1.1%)	0.377	0.43 *
	28 (29.8%)	53 (56.4%)	13 (13.8%)	0 (0.0%)	0 (0.0%)		
Explains what the coach likes to do in their leisure time	25 (26.6%)	52 (55.3%)	17 (18.1%)	0 (0.0%)	0 (0.0%)	0.511	0.34 *
	28 (29.8%)	53 (56.4%)	13 (13.8%)	0 (0.0%)	0 (0.0%)		
Compliments the athlete’s physical appearance	54 (59.3%)	31 (34.1%)	4 (4.4%)	2 (2.2%)	4 (4.2%)	0.424	0.50 *
	49 (52.7%)	39 (41.9%)	2 (2.8%)	3 (3.2%)	2 (2.1%)		
Speaks with diminutives to the athlete	62 (66.7%)	27 (29.0%)	1 (1.1%)	3 (3.2%)	1 (1.1%)	0.230	0.46 *
	59 (62.8%)	27 (28.7%)	4 (4.3%)	4 (4.3%)	1 (1.1%)		
Makes derogatory comments about women	66 (70.2%)	22 (23.4%)	3 (3.2%)	3 (3.2%)	0 (0.0%)	1.000	0.63 *
	63 (68.5%)	24 (26.1%)	3 (3.3%)	2 (2.2%)	2 (2.1%)		
Pinches the athlete	78 (84.8%)	11 (12.0%)	1 (1.1%)	2 (2.2%)	2 (2.1%)	1.000	0.35 *
	79 (84.9%)	11 (11.8%)	1 (1.1%)	2 (2.2%)	1 (1.1%)		
Massages the athlete’s back	70 (76.1%)	19 (20.7%)	0 (0.0%)	3 (3.3%)	2 (2.1%)	1.000	0.58 *
	67 (71.3%)	26 (27.7%)	0 (0.0%)	1 (1.1%)	1 (1.1%)		
Asks the athlete questions about their sex life	71 (76.3%)	17 (18.3%)	2 (2.2%)	3 (3.2%)	1 (1.1%)	1.000	0.54 *
	68 (73.1%)	23 (24.7%)	2 (2.2%)	0 (0.0%)	1 (1.1%)		
Stares at the athlete’s breasts or backside	72 (79.1%)	10 (11.0%)	3 (3.3%)	6 (6.6%)	3 (3.2%)	0.454	0.44 *
	78 (84.8%)	6 (6.5%)	3 (3.3%)	5 (5.4%)	2 (2.1%)		
Shows sexual interest in the athlete	86 (92.5%)	1 (1.1%)	2 (2.2%)	4 (4.3%)	1 (1.1%)	1.000	0.27 *
	88 (93.6%)	1 (1.1%)	2 (2.1%)	3 (3.2%)	0 (0.0%)		
Kisses the athlete on the lips	91 (96.8%)	0 (0.0%)	0 (0.0%)	3 (3.2%)	92 (96.8%)	0.500	0.59 *
	91 (97.8%)	0 (0.0%)	1 (1.1%)	1 (1.1%)	0 (0.0%)		
Proposes sex with nothing in return	91 (97.8%)	0 (0.0%)	0 (0.0%)	3 (3.2%)	92 (96.8%)	1.000	0.49 *
	91 (96.8%)	1 (1.1%)	1 (1.1%)	1 (1.1%)	0 (0.0%)		
Proposes sexual relations in exchange for privileges	91 (97.8%)	0 (0.0%)	0 (0.0%)	3 (3.2%)	93 (97.9%)	0.500	0.59 *
	92 (97.9%)	0 (0.0%)	1 (1.1%)	1 (1.1%)	0 (0.0%)		

Note: * *p* < 0.05 significant differences between the Test and the re-Test. Sometimes: Yes. Ever. Often: Yes. Often.

## Data Availability

If necessary, you can contact the authors to obtain it.

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
