# Peer review of "Assessing the Reliability of the Sexual Violence Questionnaire in Sport among Spanish-Speaking Athletes"

_ijerph, 2024, doi:10.3390/ijerph21091214_

Round 1
Reviewer 1 Report
Comments and Suggestions for Authors
Thanks for opportunity to review manuscript entitled ‘‘Assessing the Reliability of the Sexual Violence Questionnaire in Sport Among Spanish-Speaking Athletes’’ for International Journal of Environmental Research and Public Health. Overall, although the article is generally well written and deserves to be published in this journal some necessary and minor revisions must be made before the publication of the article. The strength of the manuscript includes examining variables of interest in a country where such studies are scarce and examining an interesting topic that need more scientific studies. Because my main philosophy of reviewing a manuscript as reviewer and sometimes an editor to improve the manuscript and not punishing the authors, I provided very specific and detailed peer review of the manuscript to increase its quality and citation potential. I hope author of the manuscript may benefit from my review. Necessary revisions reported section by section with the page and line number and when possible with suggestions.
Title
1. No need revision.
Abstract
2. Page 1, Line 20-21: In the following sentence ‘ ‘A sample of 146 (52 female, 94 male) Bachelor students from a university in the Basque Country participated in this cross cross-sectional study’’ Bachelor must be bachelor.
3. Page 1, Line 20-21: In the following sentence ‘‘However, Cohen’s Kappa analysis showed significant agreement across all items.’’ agreement or disagreement? However used to indicate negative situations.
4. Abstract, General: Authors need to add Keywords Sexual Violence Questionnaire and reliability as their title indicate.
5. Abstract, General: As seen in Results some Cohen’s Kappa values are very low for some items. Authors need to indicate this in abstract.
Introduction
6. Page 1, Line 39-40: Following ‘‘In a study involving of 10000 athletes from 6 European countries, it was found that 35% and 20% of athletes had experienced non-contact sexual violence and contact violence respectively before the age of 18’’ must correct as ‘ ‘In a study involving of 10000 athletes from six European countries…..’’
7. Introduction General: Some problems exist in Introduction section. First authors did not give any information about previous studies that examine reliability of Sexual Violence Questionnaire across different cultures and weaknesses that necessitate current study.
8. Introduction General: Second, authors need to give more information about importance of study. Specifically, authors need to answer Why it is important and necessary to examine reliability of the Sexual Violence Questionnaire in Sport among Spanish-Speaking athletes? I am not able to see this in Introduction section.
9. Introduction General: Third, authors also need to answer why they did not examined construct validity and solely focused on reliability.
Method
10. Method General: All small n representing subgroups must be italic in Participants section.
11. Method General: following must move to Procedure section. ‘‘Before administering the questionnaire, all participants were informed of the objectives of the study, as well as the research procedure, and the possible risks. Their participation was voluntary, and they signed an informed consent form. The study was approved by the Ethics Committee for Human Research of the University of the Basque Country (M10/2022/110).’’
12. Method General: the numbering of Material and Methods section is wrong and must be corrected: 2.2. Procedure, 2.2. Instrument (2.3. Instrument)
Results
13. Results section, General: Information in Table be must not be italic. 0.892
14. Results section, General: add Note. before explanations under the table.
Discussion
15. Discussion, general: Practical implications of study findings are completely missing and must add discussion section with a paragraph.
16. Discussion, general: Author must give possible explanations why it is low after this ‘ ‘These values determine the reliability and temporal stability of the instrument, although they are relatively lower than those obtained with the instrument validated to measure the harassment in team sports [35].’’
Conclusion
17. No problem exists in this section.
Comments on the Quality of English LanguageModerate editing of English language required
Author Response
Thank you for your detailed and constructive comments on our manuscript. We sincerely appreciate your time and effort to provide such a careful and specific review.
We are pleased that you found merit in our work, especially in the context of addressing an understudied topic in our region. We understand the importance of the revisions you have suggested to improve the quality of the manuscript and increase its impact.
Kind regards.

Reviewer 2 Report
Comments and Suggestions for Authors
The authors present a topic of great interest and of particular topicality at this time. The introduction is in line with the object of study of this paper, but the authors are advised to incorporate more current references. The same is true of the discussion. Both sections present a lower than desired percentage of references from the last five years.
The section on materials and methods is correctly described by the authors. They make use of the appropriate statistical descriptors for the type of study they have carried out and this is shown in the results section. The results are presented very clearly and in great detail, as shown in the tables they have presented.
Finally, as I have already mentioned, the discussion should be reinforced with more current references. With this small improvement, the publication of this research is recommended.
Comments on the Quality of English LanguageMinor editing of English language required.
Author Response

(The authors gave the same response as above.)

Reviewer 3 Report
Comments and Suggestions for Authors
The article aims to: this study aimed to analyze the internal consistency and reliability of the questionnaire on sexual harassment in sports, originally designed and developed in English and later adapted to Spanish/Catalan.
Address the following items:
1) The abstract section does not clearly state the research objective. It should be described before the content related to the methods.
2) Specify the type of sampling and the criteria used for its selection. Determine if the sample is representative of the population in case a probabilistic sampling method is used.
3) Specify the normality test used.
4) When studying a specific sample from the Universidad del País Vasco (case study), it is also a research limitation that prevents the generalization of results. This limitation should be specified at the end of the discussion section.
5) To fully meet the research objective, the authors should recommend further studies with diversified and representative samples.
Author Response

(The authors gave the same response as above.)

Reviewer 4 Report
Comments and Suggestions for Authors
Your study makes a valuable contribution to the field by adapting and validating the Sexual Violence Questionnaire in Sports for a Spanish-speaking population. With some revisions, particularly in the areas of literature review, methodology, and language, the manuscript will become even stronger and more impactful.
Introduction and literature review: The introduction provides a good overview of the issue of sexual violence in sports, particularly within the coach-athlete relationship. However, consider expanding the discussion on cultural differences in perceptions of sexual violence, as this is a critical aspect of this study. Including more recent studies on this topic could strengthen your argument for the need to adapt the Sexual Violence Questionnaire in sports to different cultural contexts.
Methodology: The research design was sound, but more details on the sample selection process would be beneficial. Specifically, explain how the sample was chosen and whether it was from a broader population. This would help readers better understand the generalizability of your findings.
- In the Methods section, consider clarifying the steps taken to ensure the comprehensibility of the questionnaire after translation. More details on how the pilot study informed these modifications would be valuable.
- The results are clearly presented, but some tables could benefit from additional explanatory notes or summaries to make the findings more accessible to readers who may not be familiar with statistical methods. Consider adding a brief interpretation of the key results alongside the tables. - Your discussion effectively highlights the significance of your findings, but it could be strengthened by addressing the potential limitations of your study. For example, consider discussing the potential impact of the sample's demographic characteristics on the findings and how this might influence the interpretation of the results. While your conclusion is well founded, emphasizing the practical implications of your findings for policymakers and practitioners in sports could further enhance the impact of your study.
Language and presentation: The overall quality of English is good, but phrasing can be smoother in some areas. Consider revising sentences for clarity and consistency, particularly in terms of article usage and tense consistency. Breaking up longer sentences can also improve readability. Ensure that all sections transition smoothly to maintain the flow of the manuscript. This will help make the paper easier for readers to follow.
1. Minor grammar issues: In some instances, the use of articles ("the," "a," etc.) is inconsistent or missing, which can disrupt the flow of the text. For instance, "The prevalence of sexual harassment and abuse in sport, has now been extensively documented and reported within academic research" could be better phrased as "The prevalence of sexual harassment and abuse in sport has been extensively documented and reported in academic research."
2. Phrasing and word choice: Some sentences are somewhat lengthy and can be broken down for clarity. For example, "Given the cultural differences identified in other studies and the fact that the questionnaire was initially developed in English and adapted to this cultural context, this study aimed to analyze the internal consistency and reliability of the questionnaire on sexual harassment in sports..." could be rephrased for simplicity and better flow.
3. Technical terminology: The use of technical terminology is appropriate, but in some cases, it might benefit from brief definitions or explanations, particularly for a broader audience that may not be familiar with certain statistical terms or research methodologies.
4. Consistency: Maintaining consistency in tense and voice (active vs. passive) throughout the document improves readability. For example, sometimes the text switches between the past and present tense without a clear reason.
References: The cited references are relevant and appropriate for this research. However, you might want to include a few more recent studies, particularly those that discuss cultural variations in perceptions of sexual violence. This could provide a more comprehensive background and further support your findings.
Comments on the Quality of English LanguageLanguage and presentation: The overall quality of English is good, but phrasing can be smoother in some areas. Consider revising sentences for clarity and consistency, particularly in terms of article usage and tense consistency. Breaking up longer sentences can also improve readability. Ensure that all sections transition smoothly to maintain the flow of the manuscript. This will help make the paper easier for readers to follow.
1. Minor grammar issues: In some instances, the use of articles ("the," "a," etc.) is inconsistent or missing, which can disrupt the flow of the text. For instance, "The prevalence of sexual harassment and abuse in sport, has now been extensively documented and reported within academic research" could be better phrased as "The prevalence of sexual harassment and abuse in sport has been extensively documented and reported in academic research."
2. Phrasing and word choice: Some sentences are somewhat lengthy and can be broken down for clarity. For example, "Given the cultural differences identified in other studies and the fact that the questionnaire was initially developed in English and adapted to this cultural context, this study aimed to analyze the internal consistency and reliability of the questionnaire on sexual harassment in sports..." could be rephrased for simplicity and better flow.
3. Technical terminology: The use of technical terminology is appropriate, but in some cases, it might benefit from brief definitions or explanations, particularly for a broader audience that may not be familiar with certain statistical terms or research methodologies.
4. Consistency: Maintaining consistency in tense and voice (active vs. passive) throughout the document improves readability. For example, sometimes the text switches between the past and present tense without a clear reason.
Author Response

(The authors gave the same response as above.)

Round 2
Reviewer 1 Report
Comments and Suggestions for Authors
Thanks for opportunity review revised manuscript entitled ‘‘Assessing the Reliability of the Sexual Violence Questionnaire in Sport Among Spanish-Speaking Athletes’’. I would like the thanks to authors. They make a good job for improving quality of their manuscript. Authors revised the manuscript as I requested with a good will. In this form, Introduction reflects well the previous studies and study aim, Method section and Result section is correct, and Discussion section adequately synthesis to previous study findings and current study results. Overall, I have no further comment regarding to manuscript. I congratulate to authors and wish them success on their future endeavors.
Reviewer 3 Report
Comments and Suggestions for Authors
The above issues have been resolved.